# Technical note: Accounting for snow in the estimation of root-zone water storage capacity from precipitation and evapotranspiration fluxes

David N. Dralle[1], W. Jesse Hahm[2], K. Dana Chadwick[3,4], Erica McCormick[3], and Daniella M. Rempe[3]

[1]Pacific Southwest Research Station, United States Forest Service, Davis, CA, USA
[2]Department of Geography, Simon Fraser University, Burnaby, BC, Canada
[3]Jackson School of Geosciences, University of Texas at Austin, Austin, TX, USA
[4]Department of Integrative Biology, University of Texas at Austin, Austin, TX, USA

**Correspondence:** D.N.D. (david.dralle@usda.gov)

**Abstract.** A common parameter in hydrological modeling frameworks is root-zone water storage capacity ($S_R[L]$), which mediates plant-water availability during dry periods and the partitioning of rainfall between runoff and evapotranspiration. Recently, a simple flux-tracking based approach was introduced to estimate the value of $S_R$ (Wang-Erlandsson et al., 2016). Here, we build upon this original method, which we argue may overestimate $S_R$ in snow-dominated catchments due to snow
melt and evaporation processes. We propose a simple extension to the method presented by Wang-Erlandsson et al. (2016), and show that the approach provides a lower estimate of $S_R$ in snow-dominated watersheds. This $S_R$ dataset is available at 1 km resolution for the continental United States, along with the full analysis code, on Google Colaboratory and Earth Engine platforms. We highlight differences between the original and new methods across the rain-snow transition in the Southern Sierra Nevada, California, USA. As climate warms and precipitation increasingly arrives as rain instead of snow, the subsurface may
be an increasingly important reservoir for storing plant-available water between wet and dry seasons; improved estimates of $S_R$ will therefore better clarify the future role of the subsurface as a storage reservoir that can sustain forests during seasonal dry periods and episodic drought.

## 1   Introduction

Root-zone water storage capacity ($S_R[L]$) quantifies the maximum amount of subsurface water that can be stored for use
by vegetation. This ecohydrological parameter plays a central role in the determination of plant community composition and drought resilience (Hahm et al., 2019a, b), runoff generation mechanisms (Botter et al., 2007; Salve et al., 2012), landslide triggering (Montgomery and Dietrich, 1994), landscape evolution (Deal et al., 2018), and the partitioning of precipitation into evapotranspiration and runoff (Porporato et al., 2004). Practically, *in situ* measurement of $S_R$ at large spatial scales is infeasible, leading to the development of various methods for estimating $S_R$ using remote sensing and model inversion approaches
(de Boer-Euser et al., 2016; Gao et al., 2014; Wang-Erlandsson et al., 2016; Dralle et al., 2020). Although high-resolution maps of *soil* plant-available water storage capacity exist (Reynolds et al., 2000), such maps incompletely describe the water used by plants. This may be because plants are unable to access the full reported depth of the soil column, or because plants access

water stored at depths below soil (Dawson et al., 2020; Schwinning, 2010). For example, roots can extend into and draw water from the bedrock vadose zone (rock moisture, *sensu* Rempe and Dietrich, 2018; Hahm et al., 2020) or groundwater (Miller et al., 2010; Lewis and Burgy, 1964). Within seasonally dry environments in particular, a significant volume of water accessed during the growing season can be derived from depths below mapped soils (Rose et al., 2003; Jones and Graham, 1993; Arkley, 1981). We emphasize that an accurate representation of $S_R$ therefore should include not only moisture available within the soil, but also plant-accessible water below the soil, which may include unsaturated moisture in weathered rock or groundwater.

$S_R$ does not, however, include snowpack, which is an above-ground water storage reservoir. Correctly estimating $S_R$ in systems that currently receive a significant proportion of their precipitation as snow is particularly important given the ongoing shift from snow to rain under a warming climate (e.g. Knowles et al., 2006), and the attendant heightened significance of subsurface water storage dynamics to plant ecosystems and streams. An existing, widely-used method for estimating $S_R$ (Wang-Erlandsson et al., 2016) does not account for snowpack, which we show may result in overestimation of $S_R$. Here, we present an extension to the original method to account for snow in calculating $S_R$. We describe the method details and highlight results from a rain-snow transition transect in the Southern Sierra Nevada, California, USA. We also provide a geotiff raster map of $S_R$ across the continental United States at the 1 km pixel scale. Finally, we link to a Google Earth Engine (https://earthengine.google.com/) script written in Python (https://www.python.org/) within the Colab coding environment (https://colab.research.google.com/) to document application of the method, and to facilitate comparative analyses using other widely available and spatially distributed precipitation, snowcover, and actual evapotranspiration datasets.

## 2 Method

To estimate $S_R$, Wang-Erlandsson et al. (2016) compute a running root-zone storage deficit (more positive means larger capacity in the subsurface for moisture storage) using differences between fluxes exiting ($F_{out}$) and entering ($F_{in}$) the root zone during a given time interval (typically equal to the sampling period of the remotely sensed evapotranspiration dataset). Typically, $F_{in}$ and $F_{out}$ are set equal to precipitation ($P$) and evapotranspiration ($ET$), respectively. However, to obtain a robust lower bound estimate of $S_R$, it is important to make sure that $F_{in}$ is not underestimated (when in doubt, assume all precipitation enters the rooting zone), and that $F_{out}$ is not overestimated (when in doubt on the amount of $F_{out}$ that contributes to increases in the root zone storage deficit, simply set $F_{out} = 0$). This is a general strategy also employed by Wang-Erlandsson et al. (2016). In particular, the method occasionally enforces zero values for $F_{out}$ and $F_{in}$ to ensure that deficit calculations are not over-estimated in light of uncertainty in the timing or magnitude of fluxes; this is not equivalent to assuming that these fluxes are zero. For example, Wang-Erlandsson et al. (2016) set runoff/leakage fluxes from the root zone to zero, not because runoff/leakage do not occur, but because the magnitude and timing of these fluxes are difficult to estimate with remotely sensed data products.

The original storage deficit tracking (and subsequent estimation of $S_R$) procedure presented by Wang-Erlandsson et al. (2016) is achieved through two steps. First, over a given time interval $t_n$ to $t_{n+1}$, the accumulated difference ($A_{t_n \to t_{n+1}}$) between $F_{out}$ and $F_{in}$ is calculated as:

$$A_{t_n \to t_{n+1}} = \int_{t_n}^{t_{n+1}} F_{out} - F_{in} \, dt. \tag{1}$$

Here, since the root-zone storage *deficit* is being calculated (and not actual storage), the incoming and outgoing fluxes have opposite signs from a conventional mass balance (outgoing fluxes minus incoming fluxes for deficit calculations, as opposed to incoming fluxes minus outgoing fluxes for storage). A lower bound on the root-zone storage deficit at each time interval can then be calculated as the maximum value of $0$ and the running sum of these accumulated differences:

$$D(t_{n+1}) = \max\left(0, D(t_n) + A_{t_n \to t_{n+1}}\right) \tag{2}$$

Finally, $S_R$ is estimated as the maximum observed value of $D$.

The potential inaccuracies introduced by this original method that we explore here are that, during periods when snowpack is present within the pixel, $F_{in}$ may be non-zero due to melting snow entering the rooting zone, for example, or $F_{out}$ from the root zone may be overestimated (due to attribution of sublimation/evaporation from the snow surface to a flux from the subsurface). As discussed above, both of these possibilities may lead to overestimation of $S_R$.

In the absence of spatially and temporally resolved information about snowmelt and sublimation dynamics, a simple way to correct for these potential errors is to continue to decrease the storage deficit as incoming precipitation arrives, and to set $F_{out} = 0$ during periods when snow cover ($C$, the fraction of the pixel covered in snow, which is reliably measured at large spatial scales via satellites) is present, thereby not counting evapotranspiration towards increasing the storage deficit during snowy periods. We therefore introduce a correction term for the outgoing flux in the calculation of the accumulated difference between outgoing and incoming fluxes during each time interval:

$$A_{t_n \to t_{n+1}} = \int_{t_n}^{t_{n+1}} (1 - \lceil C - C_0 \rceil) \cdot F_{out} - F_{in} \, dt \tag{3}$$

where $C_0$ is some threshold below which it is assumed that snow cover is negligible, and $\lceil \cdot \rceil$ is the ceiling operator (rounding up to the nearest integer), returning a 1 if $C > C_0$ and 0 if $C \leq C_0$. The expression therefore effectively sets $F_{out} = 0$ whenever snow is present (or deemed not-negligible) in the pixel, providing a lower-bound estimate of $S_R$ in the running storage deficit calculation.

## 2.1 Algorithm implementation and datasets

We implement the original and snow-corrected algorithm developed here using Google Earth Engine, accessed via a Google
Colab notebook and the Python programming language's Earth Engine application programming interface. This readily enables i) access to distributed timeseries of hydrological products (i.e., snowcover, evapotranspiration, and precipitation); ii)
computation in the cloud and iii) a shareable script that can be quickly modified and executed by new users (see link at end of
manuscript).

The algorithm requires precipitation and evapotranspiration datasets to compute $F_{in}$ and $F_{out}$, and a snow cover dataset to
implement the proposed snow-correction step. We use Oregon State's PRISM daily precipitation product (https://developers.
google.com/earth-engine/datasets/catalog/OREGONSTATE_PRISM_AN81d) (Daly et al., 2008, 2015), available at 2.5 arc
minute resolution. For evapotranspiration, we use the cloud-corrected Penman-Monteith-Leuning Evapotranspiration V2 product (https://developers.google.com/earth-engine/datasets/catalog/CAS_IGSNRR_PML_V2)(Zhang et al., 2019; Gan et al., 2018),
available at an 8 day timestep and 500 m resolution, and sum the vegetation transpiration, interception, and soil evaporation
bands to calculate total evapotranspiration. For the snowcover dataset, we use the Normalized Difference Snow Index (NDSI)
snow cover band from the 500m MODIS/Terra data product (https://developers.google.com/earth-engine/datasets/catalog/
MODIS_006_MOD10A1) (Hall et al., 2016), and set $C_0 = 0.1$ (snow cover is assumed negligible at less than $C_0 = 10\%$
pixel coverage; in this case, $C_0 = 10\%$ is also the minimum non-zero value of the underlying snow cover dataset). We restrict
our analysis to the temporal intersection of these three datasets (the root zone storage deficit is tracked continuously from
the 2003 to the 2017 water year), reproject into WGS84 (EPSG:4326), and resample pixels using nearest-neighbor to a 32.34
arc-second pixel scale (approximately 1 km).

We mask out pixels from our analysis where we anticipate our method will fail to accurately estimate $S_R$, namely urban areas,
open water, and croplands (which are typically subject to irrigation). To generate this mask, we use the 'LC_Type1' band from
the 2001 year of the MODIS MCD12Q1 v6 landcover product (https://developers.google.com/earth-engine/datasets/catalog/
MODIS_006_MCD12Q1) (Friedl and Sulla-Menashe, 2015). In some areas (e.g., deserts), dataset errors or unaccounted-for
inter-pixel flow result in unrealistic $S_R$ estimates, as described by Wang-Erlandsson et al. (2016). In the case that inter-pixel
flow results in a net contribution to the root-zone, estimates of $S_R$ in our (and the original) method may not represent true
lower bounds. At present, however, there are few if any methods for reliably measuring such inter-pixel fluxes at large scales,
let alone for determining whether vegetation has access to these fluxes. Wang-Erlandsson et al. (2016) suggest a potential
correction technique for this issue by adding the long term average difference $ET - P$ (where it is positive) to $F_{in}$. However,
we choose to remove these areas entirely from our data product by masking out pixels where cumulative evapotranspiration
over the study period exceeds cumulative precipitation. If needed, this correction method implemented by Wang-Erlandsson
et al. (2016) can easily be added to the code notebook published alongside this manuscript.

Finally, to provide an example of the impact of the method, we focus on the western slope of the Sierra Nevada in California, United States, where elevations range from approximately 100 m to 4000 m, driving strong gradients in mean annual
temperature (-1.5 °C to 17.5 °C, PRISM Climate Group (2017)), mean annual precipitation (120 mm to 1500 mm, PRISM

Climate Group (2017)), vegetation cover (oak savanna at low elevations to mixed-conifer forest at high elevations) and annual maximum snow cover (0% to 100%).

## 3   Results

Figure 1 illustrates three raster data layers in the Sierra Nevada focus region derived from application of the new method. Figure 1a plots root-zone storage capacity calculated using the snow-correction method. Values range from near 0 mm over exposed bedrock outcrops in the High Sierra, to over 900 mm in the dense mid-elevation forests. Figure 1b shows the difference between $S_R$ computed using the original method and the snow-corrected $S_R$. Figure 1c plots average winter (January through April) snow cover. As expected, the difference in Figure 1b is small in the lower, rain-dominated elevations, and larger in areas with snow cover. However, some areas with substantial snow cover show small differences between the methods. These are likely areas where root-zone storage capacity is small, coinciding with exposed-bedrock locations at high elevations.

Figure 2 illustrates the full timeseries output of the snow-accounting and original methods at two locations, identified by white points in Figure 1. The location farther west is a 'low snow' location, with negligible snowfall (snow present less than 1% of the time) during the winter months, and the location to the east is a 'high snow' location, with snowcover present over 50% of the time during the winter months. Gray shading in all subplots indicates that greater than 10% of the pixel is covered in snow at that time point, during which evapotranspiration is set to zero in our method (lower panels). The top panels of Figure 2 plot storage deficits using the original and the snow-accounting methods, clearly demonstrating the divergence of deficit calculations between the two methods in the region with significant snow cover. In all instances, $S_R$ is calculated as the maximum observed value of the storage deficit. In the high snow location using the original method, this leads to an estimated value of $S_R$ that is approximately 50% larger than that calculated with the snow-accounting method.

## 4   Discussion

Our proposed method for estimating $S_R$ provides a minimum estimate. Actual $S_R$ should generally exceed estimated $S_R$ values presented in our revised method, because some evapotranspiration occurs during times when snow cover is present. The snow-accounting method and the original method do not account for leakage, surface runoff, and upslope drainage in the calculation of $F_{in}$.

Drawbacks associated with the general approach are presented in detail in Wang-Erlandsson et al. (2016). In particular, the results are highly sensitive to the quality of the underlying remote-sensing datasets; by making our code publicly available, we hope that as improved datasets become available they can be readily incorporated to produce better estimates of $S_R$. As noted in a similar effort by Dralle et al. (2020), we caution against using evapotranspiration datasets which rely on a soil water balance that as a model parameter incorporate pre-determined values of $S_R$ (e.g., from existing soils databases), as this would bias the inferred $S_R$.

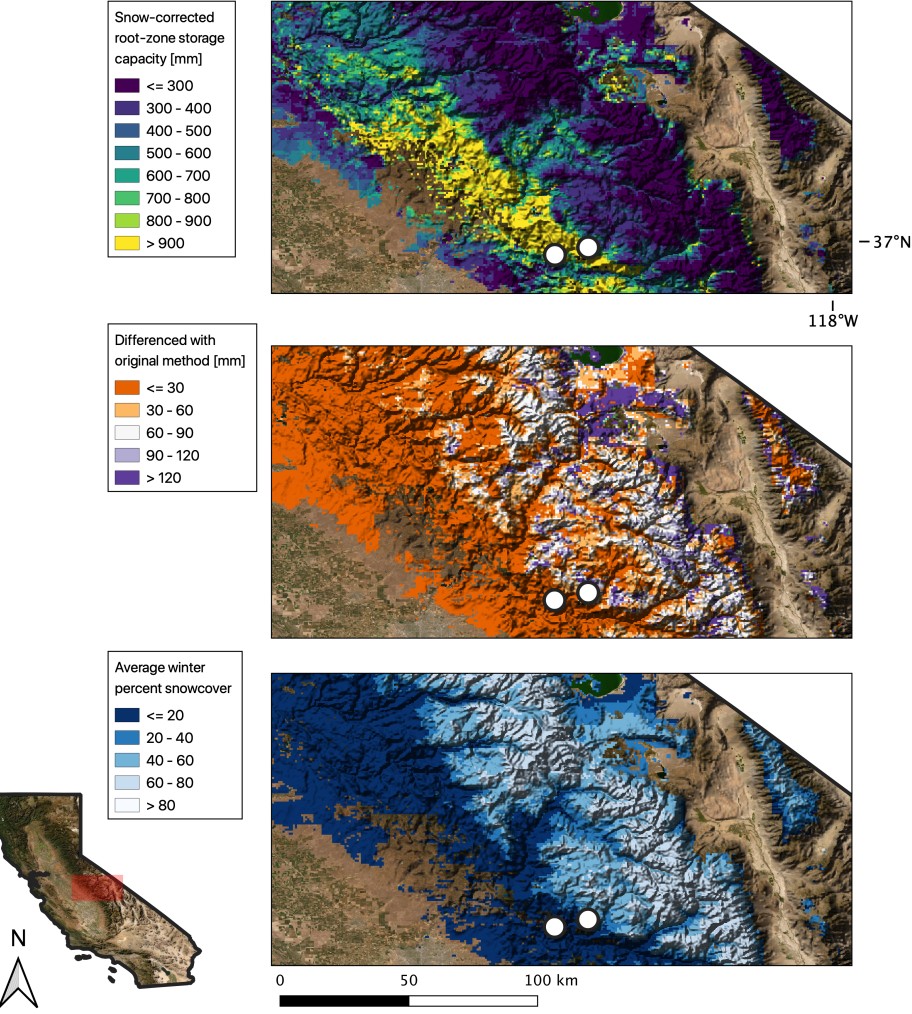

**Figure 1.** Maps of snow-corrected $S_R$ (a), the difference between the original and snow-corrected $S_R$ (b), and average winter (January - April) percent snow-cover (c) over a region of the Southern Sierra Nevada, California, USA. White points identify rain-dominated (western) and snowy (eastern) locations highlighted in Fig. 2. Imagery obtained from publicly available data through U. S. Department of Agriculture, Farm Service Agency's National Agriculture Imagery Program.

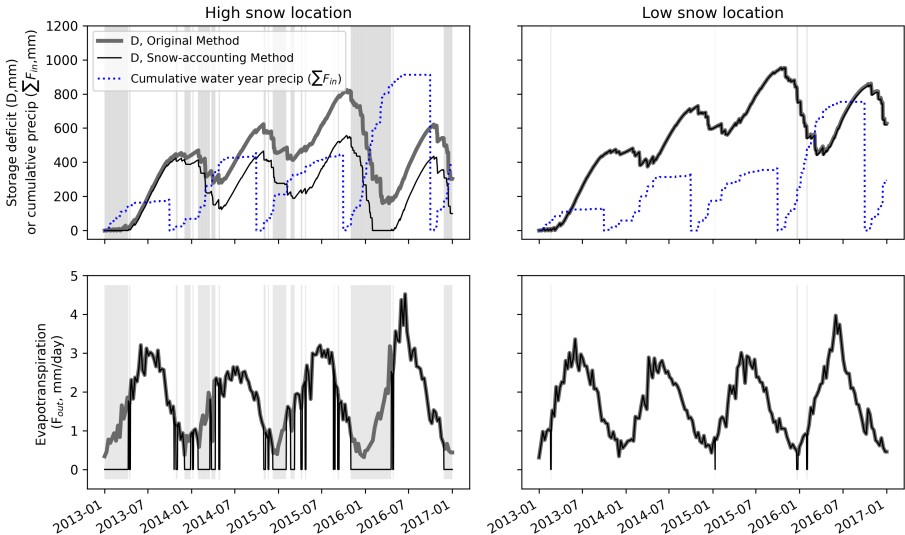

**Figure 2.** Storage deficit timeseries for representative 'high-snow' and 'low-snow' locations in the Sierra Nevada (locations mapped in Fig. 1) from 2013 through 2016, showing the difference between the original and snow-accounting methods. Cumulative water year precipitation (instantaneous precipitation is $F_{in}$) is plotted in dotted blue along the top row, and ET ($F_{out}$) is plotted in the bottom row, where the snow-accounting method takes $F_{out} = 0$ during periods of snowcover (grey shading in background). During snow-free periods (white background), deficits change identically (though there may be a vertical offset). During periods when snow is present, the new method prevents deficit growth, whereas the deficit may grow during snowy periods using the original method (e.g., Jan of 2015). The plot demonstrates how the original method may lead to a larger estimate of $S_R$ (computed as the maximum value of $D$) in snow-dominated locations.

Because the method relies on a mass balance approach, estimates of $S_R$ will inherently be larger in locations where rates of plant-water use are high during extended dry periods; for example, in the Mediterranean-type climate of California, where the long dry summer coincides with the growing season. Consequently, $S_R$ estimates will be less representative of the potentially

observable maximum of root-zone water storage capacity in wetter climates because root-zone storage deficits are frequently replenished, and therefore never reach large values. In other words, this method is only capable of documenting the root-zone storage capacity that is *accessed* by plants, rather than the *accessible* plant-available water storing capacity that may exist through the whole rooting zone (*sensu* Klos et al., 2018); the former provides a minimum estimate of the latter. In energy-limited environments, or places where seasonal precipitation and energy delivery are in phase, the method is prone to significant

underestimation of plant-accessible water.

$S_R$ in rain-dominated climates has been shown to impact drought resilience (Hahm et al., 2019a), and snow-rain transition elevations are increasing as the climate warms (Knowles et al., 2006). If precipitation arrives as rain rather than snow, the role of the subsurface in storing that water for plants will likely be amplified. Mountainous snow-rain transition zones can support high rates of ET and coincide with forested areas (Goulden et al., 2012; Hahm et al., 2014), underscoring the importance of

155 accurate estimates of $S_R$ for prediction of forest sensitivity to climate variability in the future.

Finally, we caution that neither this dataset nor the original dataset calculated by Wang-Erlandsson et al. (2016) has been validated against direct measurements of root-zone storage capacity. Although Wang-Erlandsson et al. (2016) performed an implicit validation of $S_R$ via hydrological modeling, we advocate for complementary *in situ* measurements of dynamic water storage in the critical zone, which will be required for true validation of emerging remote sensing datasets of subsurface water storage (e.g. Wang-Erlandsson et al., 2016; Enzminger et al., 2019; Swenson et al., 2003, ). Systematic validation of this form requires significant new fieldwork efforts that we leave for future work.

## 5 Conclusions

We argue that an existing method for estimating root-zone water storage capacity ($S_R$) will tend to overestimate $S_R$ in snowy areas due to unaccounted for snow melt, evaporation, and sublimation processes. We provide a correction factor that relies on a widely available distributed percent snow cover dataset to provide a tighter lower bound estimate on $S_R$. Accurately describing $S_R$ is important because the role of the subsurface in storing water is likely to be amplified in a warming climate, in which more precipitation will fall as rain rather than snow.

## 6 Data and code availability

The Python code used to implement the algorithm described here with the Google Earth Engine is available and executable as a notebook hosted on Google Colab here: https://colab.research.google.com/drive/1R6WkxaG77-O2Q7hEaiCVMvuE_1oCf_6S?usp=sharing. The datasets used to calculate $S_R$ are free and publicly accessible via the Earth Engine platform (see the links above in the Methods section above and the retrieval of the datasets within the code). The output $S_R$ raster is available at Hydroshare: https://www.hydroshare.org/resource/ee45c2f5f13042ca85bcb86bbfc9dd37/.

*Author contributions.* All authors conceived of the project. D.N.D., W.J.H., and K.D.C. wrote code, D.N.D. and W.J.H. wrote the first manuscript draft, and all authors edited the manuscript.

*Competing interests.* The authors declare no competing interests.

*Acknowledgements.* We thank Dana Lapides for helpful conversations. W.J.H. acknowledges funding support from Simon Fraser University.

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
