# Peer review of "Technical note: Accounting for snow in the estimation of root-zone water storage capacity from precipitation and evapotranspiration fluxes"

_Hydrology and Earth System Sciences, 2020_

## Author Comment (AC1) · 10 Dec 2020

Please be advised that the link to the analysis code has been updated to the following address:

https://colab.research.google.com/drive/1R6WkxaG77-O2Q7hEaiCVMvuE_1oCf_6S?usp=sharing

---

## Referee Comment (RC1) · Anonymous Referee #1 · 18 Dec 2020

Review of **Technical note: Accounting for snow in the estimation of root-zone water storage capacity from precipitation and evapotranspiration fluxes** by Dralle et al.

The manuscript of Dralle et al. deals with a new method to estimate root zone storage capacities. They based their work on the methods of Wang-Erlandsson et al., who did not account for snow. The new method adds a correction factor for snow, which leads to more conservative estimates of the root zone storage capacity in snowy areas.

I like the method and believe the manuscript is clearly written. I would like to make the authors a compliment about their open science approach. Sharing the notebook that creates the plots and links to the data is in my view an excellent example of open science, and unfortunately still rare. Nevertheless, I also have some issues, that the authors may want to address.

First, I find the discussion rather short and believe it would be good if the authors reflect a bit more thoroughly on the advantages, but especially also the disadvantages of the method. For example, the root zone estimates strongly depend on the used products for evaporation and precipitation, and the accompanying uncertainties.

In addition, I have some questions regarding C0. First, why is it chosen at 10%? This seems a bit arbitrary for me. More importantly, I wonder why the authors did not use the percentage snow cover as a correction factor by itself. I have nothing against the conservative method of the authors to just switch the correction on and off, but why not multiply Fout with 1-C? In other words, when 20% of a cell is covered in snow, then 80% of the cell can still contribute to evaporation, which is less conservative, but maybe closer to reality.

I also wonder if the necessary correction is not an artefact of the chosen evaporation product. The used soil evaporation and transpiration from the Penman-Monteith-Leuning Evapotranspiration V2 do not reach zero during winter and still reach values of 1-2 mm/d in Figure 2. However, one would expect that with snow and temperatures around zero, the transpiration and soil evaporation are zero as well. Especially as the chosen product also includes a band that accounts for snow and ice evaporation. And with zero transpiration/soil evaporation, the correction of the authors is actually unnecessary. So do you believe that with a different product, that already corrects in a better way for snowy days, this correction is actually needed? I might be worth looking at another product that uses a better correction for snow.

Furthermore, I still have some minor comments in the list below. I hope the authors find my comments useful and I look forward to a revised version of the manuscript.

**Minor comments**
P1.L21-P2L28. I fully agree here, just note that the opposite is also true: estimates of soil water storage are made for the full soil column, whereas the volume of water that roots actually use may be smaller.
P2.L31. Shift from snow to rain under a warming climate → This sounds like a statement that needs a reference.
P3.L54. I am not sure if I follow, aren't in and out always opposite of sign?
P3.L60. Due to….is zero. → I think you need to clarify this, I misunderstood first. I guess you mean that precipitation is taken as zero in the method of Wang-Erlandsson et al., whereas in reality snow melt still enters the storage. Stated like this, it looks more like an overestimation.
P3.L78. Distributed timeseries hydrological → distributed timeseries of hydrological

P3.L78. Evapotransipration → evapotranspiration

P4.L90. C0 = 10%...snow cover dataset. → what do you mean? How can a percentage be a resolution?

P4.L103. I would suggest to introduce your study site in the methods section.

P4.87-89. How did you deal with cloud cover?

P6.L128-129. Globally...forested areas → reference?

Fig.2. Maybe also add precipitation here, to have D, Fin and Fout all together.

---

## Referee Comment (RC2) · Anonymous Referee #2 · 1 Jan 2021

I reviewed the technical note "Accounting for snow in the estimation of root water storage capacity from precipitation and evapotranspiration fluxes". The paper is well written and easy to follow but lacks a significant scientific contribution in its current form. I would first like to commend the authors for the open-source nature of the analysis and dataset. The paper addresses an important need, namely to quantify root zone water storage in places receiving snow, but has several limitations that concern me with its application. As I understand it, the main contribution of the paper is to improve the Wang-Eralandsson et al. (2016) approach in places with snow. To do this the authors make three big assumptions. First that vegetation does not transpire when snow is on the ground. Second, that snow doesn't modify the timing or intensity of precipitation at

a daily time step. And third, that runoff losses are minimal (or occur only at soil saturation). We know that these assumptions are inaccurate, as I detail below. Therefore, I challenge the authors to better quantify improvement using additional observations and an uncertainty analysis. The concern that a method can be overly simple and therefore, gives erroneous results at the time and spaces scales applied (i.e. daily and 4 km) should be dispelled by the authors. To this end, I try to give constructive ways forward in three major comments and also some minor points to help improve the readability of the paper.

Major comment 1: The paper uses the justification of making conservative estimates of storage for many of the decisions. However, it is unclear to me that deeper storage estimates are in fact more conservative. One can think of many situations where shallower storage steps would be more conservative, e.g. flood prediction. Perhaps a range of possible storages would be more consistent with the information available. It doesn't matter how conservative or consistent the method is if it misinterprets important processes. To rephrase this concern, there are three major assumptions that the authors make in their formulation that need to be explored and discussed more. The first is that trees don't transpire with snow cover. That is clearly not the case in the Sierra Nevada and more care needs to be taken in justifying this assumption. Evidence of substantial transpiration prior to snow disappearance in the Sierra Nevada are shown in the following papers.

Royce, E.B., Barbour, M.G., 2001. Mediterranean climate effects. I. Conifer water use across a Sierra Nevada ecotone. Am. J. Bot. 88 (5), 911–918. https://doi.org/10.2307/2657044.

Kelly, A.E., Goulden, M.L., 2016. A montane Mediterranean climate supports year-round photosynthesis and high forest biomass. Tree Physiol. 36 (4), 459–468.

Cooper, A.E., Kirchner, J.W., Wolf, S., Lombardozzi, D.L., Sullivan, B., Tyler, S.W. and A.A. Harpold . Snowmelt causes different limitations on transpiration

in a Sierra Nevada conifer forest. Agricultural and Forest Meteorology. 291. https://doi.org/10.1016/j.agrformet.2020.108089

The second assumption is that snow doesn't modify the timing of intensity of water inputs. I find this assumption to be troubling, particularly because it is not discussed in any detail. While one can understand that this method might capture the overall/annual fluxes of water the daily dynamics are critical to how water is partitioned in the root zone (see next point). Here are some recent citations on how snow modifies water input timing and intensity:

Yan, H., Sun, N., Wigmosta, M., Skaggs, R., Hou, Z., & Leung, R. (2018). Next‐generation intensity‐duration‐frequency curves for hydrologic design in snow‐dominated environments. Water Resources Research, 54, 1093– 1108. https://doi.org/10.1002/2017WR021290

Harpold, A. A., & Kohler, M. (2017). Potential for changing extreme snowmelt and rainfall events in the mountains of the western United States. Journal of Geophysical Research: Atmospheres, 122, 13,219– 13,228. https://doi.org/10.1002/2017JD027704

My third concern is the assumption of zero outflow during snow covered periods, which is easily shown to be false with observational data. Additionally, these high elevation snow-covered zones are known to be runoff generation areas for the large downstream rivers. Consider some discussion of vertical drainage processes. It should be noted the method makes an implicit assumption that runoff is generated when the storage is full (i.e. saturation excess). Here is some relevant modeling work: Tague, C., and Peng, H. (2013), The sensitivity of forest water use to the timing of precipitation and snowmelt recharge in the California Sierra: Implications for a warming climate, J. Geophys. Res. Biogeosci., 118, 875– 887, doi:10.1002/jgrg.20073.

Hammond, J. C., Harpold, A. A., Weiss, S., & Kampf, S. K. (2019). Partitioning snowmelt and rainfall in the critical zone: effects of climate type and soil properties. Hydrology and Earth System Sciences, 23(9), 3553-3570.

The previous application of this method was in drier locations where issues of lateral and vertical subsidy of water are less important. In order to remedy these concerns, the authors need to compare their findings against observational data or somehow quantify improvement (not just difference with the previous method). The authors should consider using a recent paper (Rungee et al., 2019) in Hydrological Processes that has data for validation in the Sierra Nevada. It is the author's responsibility to show that this method reproduces some parts of reality and does not introduce artifacts via these three assumptions.

Rungee, J, Bales, R, Goulden, M. Evapotranspiration response to multiyear dry periods in the semiarid western United States. Hydrological Processes. 2019; 33: 182– 194. https://doi.org/10.1002/hyp.13322

Major comment 2: I am also concerned about the lack of uncertainty analysis given the reliance on modeled precipitation and evapotranspiration products. In particular, the authors should consider using multiple precipitation and evapotranspiration products in order to assess how much storage is sensitive to the products versus the method itself. For example, the previous work suggest that evapotranspiration uncertainty is more important than precipitation. However, these analyses were not done in wetter places like the Sierra Nevada. Precipitation products are inherently uncertain and need to be considered in the interpretation of the storage results. An easy alternative would be to use multiple precipitation products. Evapotranspiration is more challenging since the products are less validated, however, there are sufficient products to consider this or put arbitrary error terms on the existing data set. My concern is that this method portrays a certain level of sensitivity that will be specific to the climatology of the Sierra Nevada as well as the errors and uncertainties in the products themselves. Given that snowmelt strongly modifies the intensity of terrestrial water input, it's unclear to me how a non-explicit treatment of snow melt is sufficient for a paper that is trying to include snow in subsurface water storage. Again, I think some type of sensitivity analysis is needed to justify that this modulation of precipitation intensity is not a driver of soil

saturation and excess runoff (related to major concern 1).

Major comment 3: Consistency and differences with the Wang- Erlandsson et al. (2016) approach. I am a bit confused about the attempt to modify this previous work but also changing the methods. In particular, the previous work includes interception in the outgoing water flux term. Additionally, the previous work corrects for loss terms by adding back in the difference between long-term precipitation and evapotranspiration. Again the justification of a conservative estimate is used but that is unclear in terms of what that means or whether these decisions support that.

Minor comments:   I am curious about the argument on line 5: If warming scenarios show decreased snowpack and increased rain as snow, wouldn't this effectively make the Wang-Erlandsson et al. (2016) more viable over time? It seems like this proposed method is most important now and would likely diminish in its ability to represent what is happening as more precipitation shifts from snow to rain.   Lines 27-29: It would be helpful to understand how other estimates rely on only soil moisture and what sets this method apart in partitioning plant-accessible water. I'm not clear on how this marks a distinction.   Selection of fSCA threshold– Line 90: —I don't fully understand the selection criteria for the threshold below which snow contribution to the pixel isn't assumed to be meaningful. 10% seems sort of arbitrary aside from being the uncertainty for MOD10—what if that 10% is deep and holds significant water? Would be great to see more justification/explanation of how to derive this threshold since it exerts powerful control over Eq. 3. How sensitive is the result on your threshold? o Uncertainty based on the selection of a threshold—How much impact does the selection of a threshold have and if its user defined, how much would that swing the significance of the results as presented in Figure 1?; 4) Assumption about groundwater losses/contributions—how much does this assumption alter the bound if it looks fairly reasonable to assume that there are significant losses/contributions on a daily timestep in this particular region?  Enzminger, T. L., Small, E. E., & Borsa, A. A. (2019). Subsurface water dominates Sierra Nevada seasonal hydrologic storage. Geophysical Research Letters, 46, 11993– 12001. https://doi-org.unr.idm.oclc.org/10.1029/2019GL084589 • Definition of winter months –line 115: Justification of the time-period selected (January- April) for analysis. Why not expand to include Oct – June since that would be more reflective of a true winter especially with the focus on higher elevation sites? • Line 135: I'm still having a hard time grasping why this method is more important in a warming climate with more precipitation as rain. It seems like those are the exact conditions where Wang-Erlandsson is valid and where you have minimal differences (30 mm) in Figure 1. As we shift to rain dominated systems with less snow, it seems like it would reduce the importance of this contribution, not enhance it. • Line 110: Smax not previously introduced

---

## Short Comment (SC1) · 12 Jan 2021

**R1 comments:**

The manuscript of Dralle et al. deals with a new method to estimate root zone storage capacities. They based their work on the methods of Wang-Erlandsson et al., who did not account for snow. The new method adds a correction factor for snow, which leads to more conservative estimates of the root zone storage capacity in snowy areas.

I like the method and believe the manuscript is clearly written. I would like to make the authors a compliment about their open science approach. Sharing the notebook that creates the plots and links to the data is in my view an excellent example of open science, and unfortunately still rare. Nevertheless, I also have some issues, that the authors may want to address.

Thanks for your careful review and support of the efforts behind reproducible science!

First, I find the discussion rather short and believe it would be good if the authors reflect a bit more thoroughly on the advantages, but especially also the disadvantages of the method. For example, the root zone estimates strongly depend on the used products for evaporation and precipitation, and the accompanying uncertainties.

Thank you for this prompt; we propose to add discussion:

*"Drawbacks associated with the general approach are presented in detail in \citet{Wang_Erlandsson_2016}. In particular, the results are highly sensitive to the quality of the underlying remote-sensing datasets; by making our code publicly available, we hope that as improved datasets become available they can be readily incorporated into new estimates of $S_R$. As noted in a similar effort by \citet{dralle_plants_2020}, we caution against using evapotranspiration datasets which rely on a soil water balance that as a model parameter rely on pre-determined values of $S_R$ (e.g., from existing soils databases), as this would bias the inferred $S_R$.*

*Because the method relies on a mass balance approach, estimates of $S_R$ will inherently be larger in locations where rates of plant-water use are high during extended dry periods periods; for example, in the Mediterranean-type climate of California, where the long dry summer coincides with the growing season. Consequently, $S_R$ estimates will be less representative of total water storage capacity in wetter climates because root-zone storage deficits are frequently replenished, and therefore never reach large values. In other words, this method is only capable of documenting the root-zone storage capacity that is \textit{accessed} by plants, rather than the \textit{accessible} plant-available water storing capacity that may exist through the*

*whole rooting zone; the former provides a minimum estimate of the latter. In energy-limited environments, or places where seasonal precipitation and energy delivery are in phase, the method is prone to significant underestimation of plant-accessible water."*

In addition, I have some questions regarding C0. First, why is it chosen at 10%? This seems a bit arbitrary for me. More importantly, I wonder why the authors did not use the percentage snow cover as a correction factor by itself. I have nothing against the conservative method of the authors to just switch the correction on and off, but why not multiply Fout with 1-C? In other words, when 20% of a cell is covered in snow, then 80% of the cell can still contribute to evaporation, which is less conservative, but maybe closer to reality.

There is a practical reason for choosing 10%; this is the minimum non-zero value of snow-cover in the underlying data. We propose to add this comment to the manuscript:

*"(snow cover is assumed negligible at less than $C_0=10\%$ pixel coverage; in this case, $C_0=10\%$ is also the minimum non-zero value of the underlying snow cover dataset)"*

The reviewer makes a very observant point about the potential to scale Fout by 1-C, instead of using a C-threshold as a binary on/off switch for the entire pixel. We did consider this idea in formulating the approach, however, we ultimately decided not to attempt to account for sub-pixel processes, as we were unsure how the collective spectral signal of a pixel (including both snow- and non-snowy areas) that results in an ET value should be partitioned across the pixel. We hope that other future implementations of this approach improve estimates by incorporating these sub-pixel effects.

I also wonder if the necessary correction is not an artefact of the chosen evaporation product. The used soil evaporation and transpiration from the Penman-Monteith-Leuning Evapotranspiration V2 do not reach zero during winter and still reach values of 1-2 mm/d in Figure 2. However, one would expect that with snow and temperatures around zero, the transpiration and soil evaporation are zero as well. Especially as the chosen product also includes a band that accounts for snow and ice evaporation. And with zero transpiration/soil evaporation, the correction of the authors is actually unnecessary. So do you believe that with a different product, that already corrects in a better way for snowy days, this correction is actually needed? I might be worth looking at another product that uses a better correction for snow.

In California's snowy areas, including the Sierra Nevada (where we chose to highlight the method), it has been documented that ET can be far from negligible during times

with significant ground snowpack and near-freezing temperatures. For example, Goulden and Kelley (2016; doi:10.1093/treephys/tpv131) documented relatively warm (above freezing) daytime upper canopy temperatures and significant ET (via flux tower measurements) during Dec-Feb in the Sierra Nevada that coincided with deep snowpack. During spring snowmelt, daytime temperatures are even warmer, resulting in relatively high ET, while snowpack persists. Thus, the ET product we use appears to be consistent with direct observations of tree water use during periods with snowpack. We agree with the reviewer that our proposed method would be unnecessary with a dataset that does not allow for ET during times with snow cover, but do not agree that such a dataset is necessarily better or accurate.

The updated manuscript will include these additional citations:

"Mountainous snow-rain transition zones can support high rates of ET and coincide with forested areas \citep{goulden2012evapotranspiration,hahm2014bedrock}, underscoring the importance of accurate estimates of $S_R$ for prediction of forest sensitivity to climate variability in the future."

Furthermore, I still have some minor comments in the list below. I hope the authors find my comments useful and I look forward to a revised version of the manuscript.

Minor comments

P1.L21-P2L28. I fully agree here, just note that the opposite is also true: estimates of soil water storage are made for the full soil column, whereas the volume of water that roots actually use may be smaller.

We agree.

P2.L31. Shift from snow to rain under a warming climate → This sounds like a statement that needs a reference.

Thank you for pointing this out. We propose to reference Knowles et al [2006] who demonstrate this ongoing shift in precipitation phase:

Knowles, Noah, Michael D. Dettinger, and Daniel R. Cayan. "Trends in snowfall versus rainfall in the western United States." Journal of Climate 19.18 (2006): 4545-4559.

P3.L54. I am not sure if I follow, aren't in and out always opposite of sign?

We are pointing out that when tracking a storage *deficit via* change in deficit = dt*($F_{out}$ - $F_{in}$), the signs on the outgoing/incoming fluxes are the opposite from a typical storage tracking mass balance, with change in storage = dt*($F_{in}$ - $F_{out}$). We propose to add the following parenthetical to help clarify this point:

"(outgoing fluxes minus incoming fluxes for deficit calculations, as opposed to incoming fluxes minus outgoing fluxes for storage)"

P3.L60. Due to....is zero. → I think you need to clarify this, I misunderstood first. I guess you mean that precipitation is taken as zero in the method of Wang-Erlandsson et al., whereas in reality snow melt still enters the storage. Stated like this, it looks more like an overestimation.

We agree that the wording was a bit confusing. We propose to re-write as:

*The potential inaccuracies introduced by this original method that we explore here are that, during periods when snowpack is present within the pixel, $F_{in}$ may be non-zero due to melting snow entering the rooting zone, for example, or $F_{out}$ from the root zone may be overestimated (due to attribution of sublimation/evaporation from the snow surface to a flux from the subsurface).*

P3.L78. Distributed timeseries hydrological → distributed timeseries of hydrological

Thanks

P3.L78. Evapotransipration → evapotranspiration

Thanks

P4.L90. C0 = 10%...snow cover dataset. → what do you mean? How can a percentage be a resolution?

Thanks for requesting clarification, we propose to change from 'resolution' to "the minimum non-zero value".

P4.L103. I would suggest to introduce your study site in the methods section.

We propose to move the short description of the study area to the end of the methods.

P4.87-89. How did you deal with cloud cover?

The underlying PML dataset employs multiple techniques for dealing with cloud cover, including interpolation and the use of historical data for gap-filling during cloudy days.

We will note in the revised manuscript that this ET dataset is corrected for clouds. For more details, see:

Zhang, Yongqiang, et al. "Coupled estimation of 500 m and 8-day resolution global evapotranspiration and gross primary production in 2002–2017." *Remote Sensing of Environment* 222 (2019): 165-182.

P6.L128-129. Globally...forested areas → reference?

Thanks - we propose to re-write as:  "Mountainous snow-rain transition zones can support high rates of ET and coincide with forested areas \cite{gouldenEvapotranspiration along an elevation gradient in California's Sierra Nevada; Hahm2014PNAS}."

Fig.2. Maybe also add precipitation here, to have D, Fin and Fout all together.

Good suggestion, we propose to add this flux (light blue, dashed lines) to the top plot in figure 2:

---

## Short Comment (SC2) · 12 Jan 2021

**R2 comments:**

I reviewed the technical note "Accounting for snow in the estimation of root water storage capacity from precipitation and evapotranspiration fluxes". The paper is well written and easy to follow but lacks a significant scientific contribution in its current form. I would first like to commend the authors for the open-source nature of the analysis and dataset.

R2.1: Thanks!

The paper addresses an important need, namely to quantify root zone water storage in places receiving snow, but has several limitations that concern me with its application. As I understand it, the main contribution of the paper is to improve the Wang-Eralandsson et al. (2016) approach in places with snow.

R2.2: Correct, that is the goal.

To do this the authors make three big assumptions.

R2.3: Before our point-by-point responses, we want to stress that we completely agree with the reviewer on their points regarding these processes. In particular, we agree that 1) vegetation may transpire when snow is on the ground, 2) snow may alter the timing of infiltration of precipitation into the root zone, and 3) runoff/leakage losses from the rooting zone can be significant and do not necessarily occur in a threshold "field-capacity" like manner.

In our point by point responses below, we outline how our contribution does not rely on these assumptions and instead, takes an identical approach to Wang-Erlandsson et al. (2016) to arrive at a conservative estimate or lower bound on the root zone storage capacity. Generally speaking, the method occasionally enforces zero values for Fout and Fin (for example, changing Fout=ET to Fout=0 when snow is present) to ensure that deficit calculations remain conservative in light of uncertainty in the magnitude and timing of fluxes used to determine Fout and Fin. For example, Wang-Erlandsson et al (2016) set runoff/leakage fluxes from the rooting zone to zero, not because runoff/leakage do not occur, but because the magnitude and timing of these fluxes are difficult to estimate with remotely sensed data products. We hope that our responses below clarify this strategy.

First that vegetation does not transpire when snow is on the ground.

R2.4: We do not assume vegetation does not transpire when snow is on the ground. On the contrary, the potential for vegetation transpiration when snow is on the ground is precisely our concern; it is a primary reason to set Fout=0 when snow is present (we note, as described in R2.3, that setting Fout=0 is not the same as claiming ET=0 when snow is present). For if an unknown snowmelt flux enters the rooting zone as vegetation transpires, one cannot be certain whether the transpiration flux results in an increasing storage deficit, or if the deficits are being constantly replenished by snowmelt. Consequently, one might *overestimate storage deficits by letting Fout=ET when snow is present*. Since we cannot know the magnitude of snowmelt flux into the rooting zone, it is therefore a more conservative choice (with regards to calculating the root zone storage deficit) to set Fout=0 when snow is present, thereby ensuring that an artifactual deficit of the type described above cannot accrue.

The potential to overestimate the storage deficit is a general concern stated in the method originally established by Wang-Erllandson et al (2016): When determining Fout and Fin to calculate root-zone storage deficits, it is important to make sure not to overestimate fluxes leaving the rooting zone (e.g., to overcount ET) or underestimate fluxes entering the rooting zone (to undercount P). In Wang-Errlandson et al (2016),the authors were primarily concerned with overestimation of deficits due to unaccounted-for irrigation, which would have the effect of underestimating inputs into the rooting zone.

In response to the Reviewer's collection of comments regarding flux assumptions, we propose to re-write the first paragraph of the Methods section to more clearly describe the strategies employed to ensure deficit calculations remain conservative in light of uncertainty in flux timing and magnitude:

"To estimate $S_R$, \citet{Wang_Erlandsson_2016} compute a running root-zone storage deficit (more positive means larger capacity in the subsurface for storage) using differences between fluxes exiting ($F_{out}$) and entering ($F_{in}$) the root zone during a given time interval (typically equal to the sampling period of the remotely sensed evapotranspiration dataset). Typically, $F_{in}$ and $F_{out}$ are set equal to precipitation ($P$) and evapotranspiration ($ET$), respectively. However, to obtain a conservative estimate of $S_R$ (that is, a robust lower bound), it is important to make sure that $F_{in}$ is not underestimated (when in doubt, assume all precipitation enters the rooting zone), and that $F_{out}$ is not overestimated (when in doubt on the amount of $F_{out}$ that contributes to increases in the root zone storage deficit, simply set $F_{out}=0$). This is a general strategy also employed in the original method developed by \citet{Wang_Erlandsson_2016}. In particular, the method occasionally

enforces zero values for $F_{out}$ and $F_{in}$ to ensure that deficit calculations remain conservative in light of uncertainty in the timing or magnitude of fluxes; this is not equivalent to assuming that these fluxes are zero. For example, \citet{Wang_Erlandsson_2016} set runoff/leakage fluxes from the root zone to zero, not because runoff/leakage do not occur, but because the magnitude and timing of these fluxes are difficult to estimate with remotely sensed data products. "

Second, that snow doesn't modify the timing or intensity of precipitation at a daily time step.

R2.5: As stated above, we agree that snow may alter the timing/intensity of fluxes entering the rooting zone (different from the arrival of precipitation); we do not assume otherwise. Because of this potential for (generally unknown) shifts in the timing of delivery of the precipitation into the root zone, from a deficit-calculation standpoint, the most conservative choice (that is, the choice that will definitely not undercount fluxes entering the root zone, which might lead to an overestimate of the actual deficit) is to assume that all precipitation enters the rooting zone as it arrives (even if it is snow and does not immediately enter the root zone), and then to not resume accruing a deficit until all snow from that precipitation event (and potentially from others) has melted.

And third, that runoff losses are minimal (or occur only at soil saturation).

R2.6: Please see proposed new wording in R2.4, and the response below, R2.11.

We know that these assumptions are inaccurate, as I detail below. Therefore, I challenge the authors to better quantify improvement using additional observations and an uncertainty analysis. The concern that a method can be overly simple and therefore, gives erroneous results at the time and spaces scales applied (i.e. daily and 4 km) should be dispelled by the authors. To this end, I try to give constructive ways forward in three major comments and also some minor points to help improve the readability of the paper.

R2.7: We appreciate the reviewer's attempt to improve the readability of the manuscript.

Major comment 1: The paper uses the justification of making conservative estimates of storage for many of the decisions. However, it is unclear to me that deeper storage estimates are in fact more conservative. One can think of many situations where shallower storage steps would be more conservative, e.g. flood prediction.

R2.8: This is a good point. We agree that overestimating the size of the "bucket" might be problematic. As the reviewer points out, for conservative flood prediction, one might be safer erring on the side of a *smaller* bucket (thus potentially obtaining an overestimation of flood frequency and, presumably, more conservative management choices related to flood mitigation). However, we stress that our proposed method does in fact produce a smaller estimate of root-zone storage than the original method outlined by Wang-Erllandson et al (2019), and is therefore "more conservative" in the sense that the reviewer describes.

Perhaps a range of possible storages would be more consistent with the information available. It doesn't matter how conservative or consistent the method is if it misinterprets important processes. To rephrase this concern, there are three major assumptions that the authors make in their formulation that need to be explored and discussed more. The first is that trees don't transpire with snow cover. That is clearly not the case in the Sierra Nevada and more care needs to be taken in justifying this assumption. Evidence of substantial transpiration prior to snow disappearance in the Sierra Nevada are shown in the following papers. Royce, E.B., Barbour, M.G., 2001. Mediterranean climate effects. I. Conifer water use across a Sierra Nevada ecotone. Am. J. Bot. 88 (5), 911–918. https://doi.org/10.2307/2657044. Kelly, A.E., Goulden, M.L., 2016. A montane Mediterranean climate supports yearround photosynthesis and high forest biomass. Tree Physiol. 36 (4), 459–468. Cooper, A.E., Kirchner, J.W., Wolf, S., Lombardozzi, D.L., Sullivan, B., Tyler, S.W. and A.A. Harpold . Snowmelt causes different limitations on transpiration in a Sierra Nevada conifer forest. Agricultural and Forest Meteorology. 291. https://doi.org/10.1016/j.agrformet.2020.108089

R2.9: We thank the reviewer for these references. As noted in the detailed responses to the reviewer's three concerns (R2.4, R2.5, and R2.11), we do not make these assumptions.

The second assumption is that snow doesn't modify the timing of intensity of water Inputs. I find this assumption to be troubling, particularly because it is not discussed in any detail. While one can understand that this method might capture the overall/annual fluxes of water the daily dynamics are critical to how water is partitioned in the root zone (see next point). Here are some recent citations on how snow modifies water input timing and intensity: Yan, H., Sun, N., Wigmosta, M., Skaggs, R., Hou, Z., & Leung, R. (2018). Next⢠Rgeneration intensityâ ˘ A˘ Rdurationâ ˘ A˘ Rfrequency curves for hydrologic design in ˘ snowâA˘ Rdominated environments. Water Resources Research, 54, 1093– 1108. ˘ https://doi.org/10.1002/2017WR021290 Harpold, A. A., & Kohler, M. (2017). Potential for changing extreme snowmelt and rainfall events in the mountains of

the western United States. Journal of Geophysical Research: Atmospheres, 122, 13,219– 13,228. https://doi.org/10.1002/2017JD027704

R2.10: We thank the reviewer for these additional citations. As noted in R2.5, we do not make this assumption.

My third concern is the assumption of zero outflow during snow covered periods, which is easily shown to be false with observational data. Additionally, these high elevation snow-covered zones are known to be runoff generation areas for the large downstream rivers. Consider some discussion of vertical drainage processes. It should be noted the method makes an implicit assumption that runoff is generated when the storage is full (i.e. saturation excess). Here is some relevant modeling work: Tague, C., and Peng, H. (2013), The sensitivity of forest water use to the timing of precipitation and snowmelt recharge in the California Sierra: Implications for a warming climate, J. Geophys. Res. Biogeosci., 118, 875– 887, doi:10.1002/jgrg.20073. Hammond, J. C., Harpold, A. A., Weiss, S., & Kampf, S. K. (2019). Partitioning snowmelt and rainfall in the critical zone: effects of climate type and soil properties. Hydrology and Earth System Sciences, 23(9), 3553-3570.

R2.11: We believe the reviewer is referring to the requirement that the deficit cannot be negative; that is, additional precipitation does not decrease the deficit below zero. It's important to note, however, that a zero deficit with this method does not imply storage is full or that runoff occurs. Instead, a zero deficit only means that storage must be equal to or greater than the value of storage when the deficit accrual began at $t=t_0$. Whether or not there is additional "space" in the root zone for storage to accrue is not something this particular method can resolve. In this regard, our approach is identical to that of Wang-Errlandson et al. (2016); the method makes no assumptions about how or when runoff/outflow events occur.

The previous application of this method was in drier locations where issues of lateral and vertical subsidy of water are less important. In order to remedy these concerns, the authors need to compare their findings against observational data or somehow quantify improvement (not just difference with the previous method). The authors should consider using a recent paper (Rungee et al., 2019) in Hydrological Processes that has data for validation in the Sierra Nevada. It is the author's responsibility to show that this method reproduces some parts of reality and does not introduce artifacts via these three assumptions. Rungee, J, Bales, R, Goulden, M. Evapotranspiration response to multiyear dry periods in the semiarid western United States. Hydrological Processes. 2019; 33: 182– 194. https://doi.org/10.1002/hyp.13322

R2.12: Originally, the method was applied globally (see the original Wang-Erlandsson et al. (2016) manuscript) and was not constrained to dry regions. Regarding the reviewer's reference to artifacts introduced via certain methodological assumptions, please see responses R2.4, R2.5, and R2.11.

Major comment 2: I am also concerned about the lack of uncertainty analysis given the reliance on modeled precipitation and evapotranspiration products. In particular, the authors should consider using multiple precipitation and evapotranspiration products in order to assess how much storage is sensitive to the products versus the method itself. For example, the previous work suggest that evapotranspiration uncertainty is more important than precipitation. However, these analyses were not done in wetter places like the Sierra Nevada. Precipitation products are inherently uncertain and need to be considered in the interpretation of the storage results. An easy alternative would be to use multiple precipitation products. Evapotranspiration is more challenging since the products are less validated, however, there are sufficient products to consider this or put arbitrary error terms on the existing data set. My concern is that this method portrays a certain level of sensitivity that will be specific to the climatology of the Sierra Nevada as well as the errors and uncertainties in the products themselves. Given that snowmelt strongly modifies the intensity of terrestrial water input, it's unclear to me how a non-explicit treatment of snow melt is sufficient for a paper that is trying to include snow in subsurface water storage. Again, I think some type of sensitivity analysis is needed to justify that this modulation of precipitation intensity is not a driver of soil saturation and excess runoff (related to major concern 1).

R2.13
- This contribution introduces an extension to an existing method, and provides the source code to carry out the method with any E or P dataset as improved datasets become available. While we provide an example of a deficit calculation using available P and ET datasets, it is not our intention to suggest that these are the only or best datasets, and it is outside of the scope of a methods-oriented Technical Note, we believe, to identify the best datasets. For these reasons, we do not conduct sensitivity analyses with multiple datasets.
- In the original manuscript introducing this general approach, Wang-Erlandsson et al. (2016) perform a global analysis across biomes and regions, including the Sierra Nevada, as well as tropical rainforest (e.g. the Amazon). We are not sure which study constrained only to more arid regions the reviewer is referring to.
- Soil saturation and excess runoff are not explicitly treated in this manuscript or the original Wang-Erlandsson et al. manuscript. We respond to this concern in R2.11.

Major comment 3: Consistency and differences with the Wang- Erlandsson et al. (2016) approach. I am a bit confused about the attempt to modify this previous work but also changing the methods. In particular, the previous work includes interception in the outgoing water flux term.

R2.14: It is true that the Wang-Erlandsson et al. (2016) ET dataset includes interception, but they themselves acknowledge that this is a drawback of their dataset, and that it would be more accurate to use transpiration and soil evaporation only (leaving out interception) in calculation of root-zone storage deficits. This makes sense; interception fluxes should not, technically, increase storage deficits in the root zone, as interception fluxes are sourced from above-ground, not below-ground, sources. Specifically, Wang-Erlandsson et al. (2016) state, "More sophisticated two-layer surface energy balance models also have the capacity to distinguish transpiration from other forms of evaporation. This implies that local root zone storage capacity can be computed, based on transpiration fluxes, which is preferred from a bio-physical point of view (although it would require estimate of interception evaporation to calculate effective precipitation). As new evaporation data sets become available, the SR estimates can easily be updated."

In summary, as the reviewer notes, we do not include interception in our outgoing water flux term. However, we stress that this is a benefit, not a drawback, of the PML evapotranspiration dataset we used.

 Additionally, the previous work corrects for loss terms by adding back in the difference between long-term precipitation and evapotranspiration. Again the justification of a conservative estimate is used but that is unclear in terms of what that means or whether these decisions support that.

R2.15: As Wang-Erlandsson et al. (2016) note, this correction technique can be applied in places where long run ET is greater than P, which will arise either due to biases in the underlying datasets and/or significant unaccounted for fluxes into the rooting zone (e.g. irrigation or inter-basin transfers of water). As opposed to making this correction, we instead opt for a more conservative approach, leaving these areas out of our analysis entirely, as it is not possible to determine the origin of the long-term imbalance in fluxes without additional information.

Minor comments: âAˇ c I am curious about the argument on line 5: If warming sce- ´ narios show decreased snowpack and increased rain as snow, wouldn't this effectively make the Wang-Erlandsson et al. (2016) more viable over time? It seems like this

proposed method is most important now and would likely diminish in its ability to represent what is happening as more precipitation shifts from snow to rain.

R2.16: The reviewer is correct: Without snow, our method would be unnecessary as it would reduce to the original method of Wang-Erlandsson et al. However, our hope is to improve estimates of root zone storage capacity now, as we are concerned with what stresses these systems are under currently. Moreover, these estimates of "true" subsurface S_R might help us (in a modeling context, for example) to better predict how snow-dominated systems might cope with only subsurface storage in a future with decreased snowpack.

Lines 27-29: It would be helpful to understand how other estimates rely on only soil moisture and what sets this method apart in partitioning plant-accessible water. I'm not clear on how this marks a distinction.

R2.17: We only intend to point out that most modeling and analysis approaches only account for plant water storage in soils. The distinction here is that we do not constrain our analysis to upper soil layers, typically within the upper 1.5 m of the subsurface in most available soils datasets (e.g. gNATSGO).

Selection of fSCA threshold– Line ´ 90: I don't fully understand the selection criteria for the threshold below which snow contribution to the pixel isn't assumed to be meaningful. 10% seems sort of arbitrary aside from being the uncertainty for MODIS. What if that 10% is deep and holds significant water? Would be great to see more justification/explanation of how to derive this threshold since it exerts powerful control over Eq. 3. How sensitive is the result on your threshold? Uncertainty based on the selection of a threshold. How much impact does the selection of a threshold have and if its user defined, how much would that swing the significance of the results as presented in Figure 1?

R2.18: The threshold is chosen primarily as a function of the underlying snow cover dataset, which has a precision of 10%. Like all other choices in this manuscript, we have opted for the conservative choice setting the threshold to the lowest non-zero value available in the dataset. In this way, our method represents an end-member case.

 4) Assumption about groundwater losses/contributions how much does this assumption alter the bound if it looks fairly reasonable to assume that there are significant losses/contributions on a daily timestep in this particular region? Enzminger, T. L., Small, E. E., & Borsa, A. A. (2019). Subsurface water domi- nates Sierra Nevada

seasonal hydrologic storage. Geophysical Research Letters, 46, 11993– 12001.
https://doi-org.unr.idm.oclc.org/10.1029/2019GL084589

R2.19: This is a reasonable concern, but it is beyond the scope of this technical note (that is concerned with the role of snow), which extends the method by Wang-Erlandsson et al. (2016) that is subject to the same problems if lateral transport is a major factor. Proper exploration of this issue would require estimates of inter-pixel transfers of groundwater.

Definition ´ of winter months –line 115: Justification of the time-period selected (January- April) for analysis. Why not expand to include Oct – June since that would be more reflective of a true winter especially with the focus on higher elevation sites?

R2.20: We do not constrain analysis to this time period alone (as noted in the manuscript, it is run continuously from 2003 to 2017). The choice to show average snowpack in January-April is purely for the purposes of illustration, to demonstrate general patterns of snow cover that might be expected to impact the underlying calculations of the method.

Line 135: I'm still ´ having a hard time grasping why this method is more important in a warming climate with more precipitation as rain. It seems like those are the exact conditions where Wang-Erlandsson is valid and where you have minimal differences (30 mm) in Figure 1. As we shift to rain dominated systems with less snow, it seems like it would reduce the importance of this contribution, not enhance it. ' c Line 110: Smax not previously ´ introduced

R2.21: Please see our explanation in R2.16.

---

## Referee Comment (RC3) · Anonymous Referee #3 · 24 Jan 2021

**General comments**

The manuscript by Dralle et al. aims to account for moisture availability in snow-dominated catchments due to snow-melting and sublimation processes by modifying the Wang-Erlandsson et al. (2016) root-zone storage capacity ($S_r$) framework. The modified framework aims to provide a more conservative $S_r$ estimate in snow-dominated catchments and is analyzed at a much finer-resolution of 1km for Southern Sierra Nevada, CA, USA.

The modification to the original framework addresses an important aspect: excess moisture availability in snow-dominated catchments, which can influence moisture

availability in a warmer climate. The manuscript is generally well written and the open access approach is laudable. However, I do have some major concerns:

- While it is clear that the modified framework provides a more conservative $S_r$ estimate (i.e., lower bound), the manuscript does *not* provide evidence that modifications also yield more accurate estimates. However, in several places in the manuscript (e.g., P6L129), it is implied that the new estimate is also more accurate. Ideally, I would suggest that the authors provide validation through e.g., hydrological modelling (with and without modified $S_r$) and validation against observation-based evaporation data or gauged runoff data. However, if providing such evidence is not within the scope of Technical notes, I would suggest that the authors make it clearer in the manuscript that there is no evidence at this point that the more conservative estimate is also more accurate.

- The term "conservative" may be confusing, as a low $S_r$ might be more conservative in certain applications (e.g., flood prediction) and less conservative in others (e.g., ecosystem service valuation of drought buffering capacity). Simply sticking to the terms like "lower-bound" or "minimum" would be less ambiguous.

- Ignoring horizontal inter-pixel flows (leakage and runoff) following Wang-Erlandsson et al. 's (2016) methodology (implemented globally at 0.5 degree resolution) at a 1km resolution for the present 'high elevation' study area can be problematic and non-conservative. Dralle et al., states (P2L46-49) that leakage and runoff are ignored, which "results in a conservative estimate of $S_r$". However, while this is true for high-elevation pixels, low-elevation pixels can expect an underestimation of $F_{in}$, and hence an overestimation of $S_r$. It is not clear to me if and how the authors address this, please clarify.

- The authors exclude the interception evaporation term from $F_{out}$ (L86), but uses total precipitation (rather than effective precipitation) for $F_{in}$. If interception evaporation is excluded, it would make sense to also exclude the non-effective precipitation, which does not interfere with sub-surface processes. While it makes $S_r$ estimates lower, it might not be for the right reasons. Or do the authors by the phrase "interception is not included" mean that both interception and non-effective precipitation are removed? If that is the case, the sentence formulation needs to be less ambiguous, especially as the term "interception" comes directly after "transpiration" and "soil evaporation".

- A suggestion for a better overview could be to introduce a table with two columns for "before" and "after" your modifications: i.e., the first column lists the Wang-Erlandsson et al original equations, and the second column lists the modified version. You could list all differences in this table, incl. for example resolution, and definitions of $F_{out}$.

- In general, it would be helpful if the authors could more systematically describe when and how the water balance is violated.

**Specific comments**
L28: '...plant-accessible water below the soil'. Does this include groundwater? Please be specific.

L49: "$F_{in}$ and $F_{out}$ are set equal to precipitation (P) and evapotranspiration (ET), respectively". However, later at L86, it is stated that "interception is not included". This can be confusing as interception evaporation generally is considered part of ET. To minimize confusion, please consider defining the $F_{in}$ and $F_{out}$ clearly once and then consistently throughout.

L53,57: 'n' for Eq. 1 and 2 are not mentioned for the $S_r$ calculation. Is the simulation run for the whole term (2003-2017), or is it simulated annually?

L55: 'root-zone storage deficit'. Suggest be consistent in terminology with Wang-Erlandsson et al., 2016.

L83: Dralle et al. have used PML-v2 evaporation product, which does a lot of plant function type (PFT) parameterization in evaporation calculation, leading to biome-based assumptions. Though, we believe that at such a fine-resolution, it shouldn't matter much. However, it would add robustness to the framework if a sensitivity analysis using a different evaporation product (e.g., FLUXCOM) can be done using the modified framework. (I would recommend this for normal articles, but acknowledge that I am not sure about the scope of "Technical notes" - maybe the editor can help provide some guidance here.)

L89-90. What is the rationale for $C_0$ = 10 %? Is $C_0$ resolution/scale/context dependent? What are your recommendations for users attempting to apply the modified algorithm on a dataset with different topography, climate, and resolution? Furthermore, the statement '$C_0$ = 10% is also the resolution of the underlying snow cover dataset' is unclear.

L110: What does $S_{max}$ represent, since it hasn't been mentioned before? What is a low-energy location? Please be more descriptive.

Fig 2. Evapotranspiration is referred to as $F_{in}$, instead of $F_{out}$.

Please be consistent with the notations for 'Root-zone water storage capacity' ($S_r$ or $S_r$[L] or $S_{max}$).

---

## Author Comment (AC2) · 2 Feb 2021

The manuscript by Dralle et al. aims to account for moisture availability in snow dominated catchments due to snow-melting and sublimation processes by modifying the Wang-Erlandsson et al. (2016) root-zone storage capacity (Sr) framework. The modified framework aims to provide a more conservative Sr estimate in snow dominated catchments and is analyzed at a much finer-resolution of 1km for Southern Sierra Nevada, CA, USA.

The modification to the original framework addresses an important aspect: excess moisture availability in snow-dominated catchments, which can influence moisture availability in a warmer climate. The manuscript is generally well written and the open access approach is laudable.

R3.1: We thank the reviewer for the careful and thorough review of the manuscript.

However, I do have some major concerns:

While it is clear that the modified framework provides a more conservative Sr estimate (i.e., lower bound), the manuscript does not provide evidence that modifications also yield more accurate estimates. However, in several places in the manuscript (e.g., P6L129), it is implied that the new estimate is also more accurate. Ideally, I would suggest that the authors provide validation through e.g., hydrological modelling (with and without modified Sr) and validation against observation-based evaporation data or gauged runoff data. However, if providing such evidence is not within the scope of Technical notes, I would suggest that the authors make it clearer in the manuscript that there is no evidence at this point that the more conservative estimate is also more accurate.

R3.2: Thanks for this insightful comment. We agree with this sentiment. We propose modifying the manuscript with:

We caution that neither this dataset nor the original dataset has been validated against direct measurements of root-zone storage capacity. Although \citet{Wang_Erlandsson_2016} performed an implicit validation of S_R via hydrological modeling, we advocate for complementary *in situ* measurements of dynamic water storage in the critical zone, which will be required for true validation of emerging remote sensing datasets of subsurface water storage \citep[e.g.][ ]{Wang_Erlandsson_2016, Enzminger-2019, Swenson-2003}. Systematic validation of this form requires significant new fieldwork efforts that we leave for future work.

• The term "conservative" may be confusing, as a low Sr might be more conservative in certain applications (e.g., flood prediction) and less conservative in others (e.g., ecosystem service valuation of drought buffering capacity). Simply sticking to the terms like "lower-bound" or "minimum" would be less ambiguous.

R3.3: We agree. We propose to use "lower-bound" throughout the manuscript, as the reviewer suggests.

• Ignoring horizontal inter-pixel flows (leakage and runoff) following WangErlandsson et al. 's (2016) methodology (implemented globally at 0.5 degree resolution) at a 1km resolution for the present 'high elevation' study area can be problematic and non-conservative. Dralle et al., states (P2L46-49) that leakage and runoff are ignored, which "results in a conservative estimate of Sr". However, while this is true for high-elevation pixels, low-elevation pixels can expect an underestimation of Fin, and hence an overestimation of Sr. It is not clear to me if and how the authors address this, please clarify.

R3.4: This is a good point, and a limitation inherent to both our proposed modification and the original method. We propose to add additional text to the manuscript:

In the case that inter-pixel flow results in a net contribution to the root-zone, estimates of S_R in our (and the original) method may not represent true lower bounds. At present, however, there are few if any methods for reliably measuring such inter-pixel fluxes at large scales, let alone for determining whether vegetation have access to these fluxes.

• The authors exclude the interception evaporation term from Fout (L86), but uses total precipitation (rather than effective precipitation) for Fin. If interception evaporation is excluded, it would make sense to also exclude the non-effective precipitation, which does not interfere with sub-surface processes. While it makes Sr estimates lower, it might not be for the right reasons. Or do the authors by the phrase "interception is not included" mean that both interception and noneffective precipitation are removed? If that is the case, the sentence formulation needs to be less ambiguous, especially as the term "interception" comes directly after "transpiration" and "soil evaporation".

R3.5: We did not include interception in the original F_out term, but agree that this might decrease the lower bound estimate of S_R, possibly for the "wrong" reasons, as the reviewer suggests. We propose to include interception in F_out as the reviewer recommends. A preliminary analysis shows this does not significantly alter observed differences with the original method, as the increase is roughly the same between both methods when interception is included. We propose to add the following to clarify the role of interception:

Although interception is not strictly sourced from the subsurface, it nevertheless may decrease the effective precipitation that reaches the subsurface. Following \citet{Wang_Erlandsson_2016} we therefore leave the interception component of the ET flux from the PML_V2 dataset in our calculation of Fout.

• A suggestion for a better overview could be to introduce a table with two columns for "before" and "after" your modifications: i.e., the first column lists the WangErlandsson et al original equations, and the second column lists the modified version. You could list all differences in this table, incl. for example resolution, and definitions of Fout.

R3.6: Because we now include interception (R3.5), the difference between the methods boils down to the difference between inclusion (or not) of snowcover in Equation 3. We don't feel we

need a table to illustrate this difference. Pixel resolution is a function of underlying choice of datasets, rather than methodology, which is the focus of this tech note.

• In general, it would be helpful if the authors could more systematically describe when and how the water balance is violated.

We propose to add the following:

The water balance is violated when F_out exceeds F_in over long time periods such that changes in storage may be considered negligible relative to cumulative sums of fluxes. This could arise, for example, due to errors in the underlying datasets, or unaccounted for input fluxes (e.g., irrigation subsidies or inter-pixel flow).

Specific comments

L28: '...plant-accessible water below the soil'. Does this include groundwater? Please be specific.

R3.6: We propose:

"We emphasize that an accurate representation of $S_R$ therefore should include not only moisture available within the soil, but also plant-accessible water below the soil, which may include unsaturated storage in weathered bedrock or groundwater."

L49: "Fin and Fout are set equal to precipitation (P) and evapotranspiration (ET), respectively". However, later at L86, it is stated that "interception is not included". This can be confusing as interception evaporation generally is considered part of ET. To minimize confusion, please consider defining the Fin and Fout clearly once and then consistently throughout.

R3.7: Thank you. Please see R3.5 for more information.

L53,57: 'n' for Eq. 1 and 2 are not mentioned for the Sr calculation. Is the simulation run for the whole term (2003-2017), or is it simulated annually?

R3.8: It is run for the whole term. We propose to further clarify in the methods:

"We restrict our analysis to the temporal intersection of these three datasets (the root zone moisture deficit is tracked continuously from the 2003 to the 2017 water year), reproject into WGS84 (EPSG:4326), and resample pixels using nearest-neighbor to a 32.34 arc-second pixel scale (approximately 1 km). "

L55: 'root-zone storage deficit'. Suggest be consistent in terminology with WangErlandsson et al., 2016.

R3.9: Wang Erlandsson et al. use various terms throughout their paper, including 'storage deficit' and 'soil moisture deficit'. We prefer storage deficit, as F_out may come from the soil or underlying weathered bedrock.

L83: Dralle et al. have used PML-v2 evaporation product, which does a lot of plant function type (PFT) parameterization in evaporation calculation, leading to biome-based assumptions. Though, we believe that at such a fine-resolution, it shouldn't matter much. However, it would add robustness to the framework if a sensitivity analysis using a different evaporation product (e.g., FLUXCOM) can be done using the modified framework. (I would recommend this for normal articles, but acknowledge that I am not sure about the scope of "Technical notes" - maybe the editor can help provide some guidance here.)

R3.11: We provide an interactive Python notebook, which can be straightforwardly edited to try different datasets. We maintain that a full inter-comparison of ET data products is beyond the scope of a methods-oriented Technical Note.

L89-90. What is the rationale for C0 = 10 %? Is C0 resolution/scale/context dependent? What are your recommendations for users attempting to apply the modified algorithm on a dataset with different topography, climate, and resolution? Furthermore, the statement 'C0 = 10% is also the resolution of the underlying snow cover dataset' is unclear.

R3.12: Please see our third comment to Reviewer 1; this is the minimum non-zero value of snow cover.

L110: What does Smax represent, since it hasn't been mentioned before? What is a low-energy location? Please be more descriptive. Fig 2. Evapotranspiration is referred to as Fin, instead of Fout. Please be consistent with the notations for 'Root-zone water storage capacity' (Sr or Sr[L] or Smax).

R3.13: Thanks, this is a typo we have fixed. We propose to remove "low-energy" (though we meant it in the sense of Budyko).

---

## Referee Report (RR1)

Review of **Technical note: Accounting for snow in the estimation of root-zone water storage capacity from precipitation and evapotranspiration fluxes** by Dralle et al.

The revised manuscript of Dralle et al., that deals with a correction for snow for estimating root-zone storage capacity with the method of Wang-Erlandsson et al. (2016), shows many improvements in comparison with the previous version. I am happy the authors found my comments useful and addressed all of them.

I especially appreciate the extended discussion and believe this addresses the possible shortcomings much better. I just have several minor issues left, from which I hope they are helpful again.

**Minor comments**
P1.L14-P2.L28. The authors agreed with me in their response that the opposite is also true: estimates of soil water storage are made for the full soil column, whereas the volume of water that
roots actually use may be smaller. Maybe it is good to also add some lines about that in the introduction?
P4.L100. Unaccounted for inter-pixel → unaccounted inter-pixel?
P4.L104. Have access → has access
P4.L109-110. I think a bit more detail on the study area would be nice.
P5.L140. Dry periods periods → dry periods
P6.L141-146. I find this paragraph rather confusing. You are comparing also now the root zone storage capacity with the total storage capacity, which are, in my view, totally different things. Also the term accessible plant-available water storing capacity is confusing, but I think also here you mean the total storage capacity in the soil. I think you should make clearly the distinction between the water that can be stored in the soils, based on soil depths and field capacities, and the root zone storage capacity, that may be less as roots do not explore the full soil column (or the top layers that are used to calculate this storage capacity).
P7.L154-155. We advocate...critical zone. → Yes, me too, but this is practically impossible on a large scale. Or do you have some ideas here?

---

## Author Response (AR2)

Pacific Southwest Research Station United States Forest Service 1731 Research Park Dr., Davis, CA 95618

Apr 26, 2021

To the Editors of Hydrology and Earth System Sciences:

Please consider for publication as a Technical Note in *Hydrology and Earth System Sciences* the revised manuscript:

**Technical note: Accounting for snow in the estimation of root-zone water storage capacity from precipitation and evapotranspiration fluxes**

Below we have included our response to the reviewer; black font represents reviewer comments, blue font our responses, and red font text that we added to the manuscript. The most significant change in this revision is our addition of additional hydroclimatic data describing the case study region in the southern Sierra.

Thank you for your consideration of this updated manuscript.

Sincerely, David N. Dralle, W. Jesse Hahm, Dana Chadwick, Erica McCormick, Daniella M. Rempe

**R1 comments:**

The revised manuscript of Dralle et al., that deals with a correction for snow for estimating root-zone storage capacity with the method of Wang-Erlandsson et al. (2016), shows many improvements in comparison with the previous version. I am happy the authors found my comments useful and addressed all of them.

I especially appreciate the extended discussion and believe this addresses the possible shortcomings much better. I just have several minor issues left, from which I hope they are helpful again:

**Minor comments**

P1.L14-P2.L28. The authors agreed with me in their response that the opposite is also true: estimates of soil water storage are made for the full soil column, whereas the volume of water that roots actually use may be smaller. Maybe it is good to also add some lines about that in the introduction?

**We edited this to include:**

This may be because plants are unable to access the full reported depth of the soil column, or because plants access water stored at depths below soil

P4.L100. Unaccounted for inter-pixel  $\rightarrow$  unaccounted inter-pixel? P4.L104. Have access  $\rightarrow$  has access

**Thanks**

P4.L109-110. I think a bit more detail on the study area would be nice. P5.L140. Dry periods periods  $\rightarrow$  dry periods

We have included additional description of the regional hydroclimate and vegetation cover.

P6.L141-146. I find this paragraph rather confusing. You are comparing also now the root zone storage capacity with the total storage capacity, which are, in my view, totally different things. Also the term accessible plant-available water storing capacity is confusing, but I think also here you mean the total storage capacity in the soil. I think you should make clearly the distinction between the water that can be stored in the soils, based on soil depths and field capacities, and the root zone storage capacity, that may be less as roots do not explore the full soil column (or the top layers that are used to calculate this storage capacity).

Thank you, we streamlined the language, and now refer to Klos et al [2018] to clarify usage of terms.

Klos, P. Zion, et al. "Subsurface plant-accessible water in mountain ecosystems with a Mediterranean climate." Wiley Interdisciplinary Reviews: Water 5.3 (2018): e1277.

P7.L154-155. We advocate...critical zone.  $\rightarrow$  Yes, me too, but this is practically impossible on a large scale. Or do you have some ideas here?

We have some ideas, but will explore them in future work!